# Anticancer Activity of Bitter Melon-Derived Vesicles Extract against Breast Cancer

**DOI:** 10.3390/cells12060824

**Published:** 2023-03-07

**Authors:** Ting Feng, Yilin Wan, Bin Dai, Yanlei Liu

**Affiliations:** Institute of Nano Biomedicine and Engineering, Shanghai Engineering Research Center for Intelligent Diagnosis and Treatment Instrument, Department of Instrument Science and Engineering, School of Electronic Information and Electrical Engineering, Shanghai Jiao Tong University, Shanghai 200240, China

**Keywords:** bitter melon, medicinal and edible plants, vesicles extract, antitumor, breast cancer

## Abstract

Due to their low immunogenicity, high biocompatibility and ready availability in large quantities, plant-derived vesicles extracts have attracted considerable interest as a novel nanomaterial in tumor therapy. Bitter melon, a medicinal and edible plant, has been reported to exhibit excellent antitumor effects. It is well-documented that breast cancer gravely endangers women’s health, and more effective therapeutic agents must be urgently explored. Therefore, we investigated whether bitter melon-derived vesicles extract (BMVE) has antitumor activity against breast cancer. Ultracentrifugation was used to isolate BMVE with a typical “cup-shaped” structure and an average size of approximately 147 nm from bitter melon juice. The experimental outcomes indicate that 4T1 breast cancer cells could efficiently internalize BMVE, which shows apparent anti-proliferative and migration-inhibiting effects. In addition, BMVE also possesses apoptosis-inducing effects on breast cancer cells, which were achieved by stimulating the production of reactive oxygen species (ROS) and disrupting mitochondrial function. Furthermore, BMVE could dramatically inhibit tumor growth in vivo with negligible adverse effects. In conclusion, BMVE exhibits a pronounced antitumor effect on 4T1 breast cancer cells, which has great potential for use in tumor therapy.

## 1. Introduction

Extracellular vesicles (EVs) are cell-derived membrane structures carrying proteins and nucleic acids as cargo [1] which play an essential role in intercellular communication and functional transduction and can be released by nearly all cell types under physiological and pathological conditions [2,3]. EVs are natural nanoparticles with low immunogenicity and a more outstanding biocompatibility and safety than other nanoparticles, which can pass through the blood-brain barrier [4] but not the placental barrier [5]. Animal-derived EVs (ADEVs), which mostly function as delivery vehicles [6], have been widely used in biomedical fields in recent years, including diagnosis, disease progression and drug delivery [7]. Moreover, plant cells also secrete EVs and participate in various physiological processes, such as defense responses [8]. In terms of application, plant-derived EVs have the unique advantage of antitumor effects and being isolated on a large scale compared with ADEVs [9,10].

According to epidemiological observations, the consumption of plant-based foods is inversely related to the occurrence of cancer [11]. The antitumor potential of plant compounds or water extracts has been investigated in previous studies [12]. At present, the antitumor activity of plant-derived EVs is gaining more attention. Since plants inevitably release some intracellular vesicles into the juice by disrupting the cells during juicing, and some intracellular vesicles are possibly mixed in the process of isolating EVs, we named the extract as plant-derived vesicles extracts (PDVEs). Ginseng-derived vesicles extract can induce apoptosis in mouse melanoma cells by promoting the polarization of the macrophages phenotype M2 to M1 and the production of reactive oxygen species (ROS) [10]. Ginger [13], garlic [14] and edible tea flowers [15] -derived vesicles extract have also been reported to have antitumor activity.

Bitter melon is a medicinal and edible plant used as a folk remedy with antitumor and anti-inflammatory effects [16]. In vitro and in vivo studies have shown that bitter melon extract (BME) has antitumor effects on oral squamous cell carcinoma (OSCC) cells [17]. BME is obtained by lyophilizing bitter melon juice supernatant, which contains several components [18]. BME can also inhibit head and neck squamous cell carcinoma (HNSCC) cell proliferation by modulating multiple signaling pathways [19]. Furthermore, BME possesses immunomodulatory effects that modulate the regulatory T cell (Treg) population in HNSCC and augment the natural killer cell-mediated killing activity of HNSCC [20]. However, studies on the antitumor activity of vesicles extract obtained from the bitter melon juice supernatant are still scarce, which only reported the antitumor activity against OSCC [21] and glioma [22]. Breast cancer is a malignant tumor that seriously threatens women’s health worldwide [23], and the role of bitter melon-derived vesicles extract (BMVE) on breast cancer cells is still unknown. We evaluated the behavior of BMVE against breast cancer using 4T1 cells as a model.

Herein, BMVE was first isolated and characterized from bitter melon juice by ultracentrifugation (Figure 1a). Then, we examined the internalization, anti-proliferation and migration inhibition effects of BMVE on breast cancer cells in vitro. Importantly, BMVE exerted a robust apoptosis-inducing effect on 4T1 cells via mitochondrial damage mediated by ROS. In addition, the 4T1 mouse model of breast cancer was created to study the biodistribution of BMVE in vivo. Finally, we confirmed that BMVE exhibited a significant antitumor effect in 4T1 tumor-bearing mice (Figure 1b).

## 2. Materials and Methods

### 2.1. Cell Line Culture

4T1 and MCF-7 cells were cultured in an RPMI 1640 medium with 10% fetal blood serum (FBS) and 1% pen/strep. GIBCO (Grand Island, NY, USA) was the supplier of cell culture products. MCF-10A cells were cultured in a dedicated medium (Procell, Wuhan, China). The incubator was maintained at 37 °C, 5% CO_2_ and under sterile conditions. The American Type Culture Collection (ATCC, Manassas, VA, USA) was the cell lines’ source.

### 2.2. BMVE Isolation

The acquired bitter melon was washed with deionized water and juiced using a juicer. The juice was centrifuged at 3000× *g* for 30 min, 10,000× *g* for 40 min at 4 °C (Centrifuge 5804 R, Eppendorf, Hamburg, Germany). After two centrifugations, the color of the supernatant changed from green to transparent. A 0.22 μm filter was used to remove the remaining impurities from the supernatant. Then, 1 mL Opti-prep was added to each tube’s bottom. The supernatant was centrifuged at 100,000× *g* for 90 min at 4 °C (Beckman Optima L-100XP, Beckman, Brea, CA, USA), and a 1 mL sample was collected from the tube’s bottom. All collected liquid was combined and ultracentrifuged under the same conditions. The pellets were suspended in sterile phosphate-buffered saline (PBS), and the resulting BMVE was frozen at −80 °C for future use.

### 2.3. BMVE Characterization

The particle size and number of bitter melon vesicles resuspended in PBS were analyzed by nanoparticle tracking analysis (NTA) (Malvern Nano As, Malvern, England), repeated 5 times. A carbon-coated mesh was dripped with five microliters of the BMVE solution and allowed to settle for one minute. Then, the excess liquid was blotted with filter paper, and the sample was stained with tungstophosphoric acid for 1 min, blotted again with filter paper and dried. BMVE was finally visualized using transmission electron microscopy (TEM) (Tecnai G2, FEI, Hillsboro, FL, USA).

### 2.4. Cell Internalization

In a four-chamber dish, 4T1 cells were seeded. The cells were treated with 20 μg/mL Dil (MKBio, Shanghai, China) -labeled BMVE for 6, 24 and 48 h. The cells were then stained for 5 min with 5 μg/mL DAPI (Beyotime, Shanghai, China) and observed using a confocal laser scanning microscope (Leica SP8 STED 3X, Wetzlar, Germany).

A 12-well plate was seeded with 4T1 cells. Cells were treated with Dil-labeled BMVE at concentrations of 5, 10 and 20 μg/mL for 6, 12, 24 and 48 h. The collected cells were then digested and analyzed by flow cytometry (BD bioscience, Franklin Lakes, NJ, USA).

### 2.5. Viability Assay

Using the Cell Counting Kit-8 (CCK-8) (Biosharp, Shanghai, China) assay, cell viability was determined. In 96-well plates, 4T1 or MCF-7 cells were seeded and exposed to various doses of BMVE for 12, 24 and 48 h. At the end of the processing time, the medium was replaced in each well and 10% CCK-8 agent was added. Cell proliferation was estimated by a microplate reader (absorbance wavelength: 450 nm).

### 2.6. Plate Colony Formation Assay

A total of 200 4T1 cells were seeded in 6-well plates. Ten days were spent cultivating cells in BMVE-containing medium. After discarding the medium and fixing the cells with methanol, the cells were stained with 0.1% crystal violet.

### 2.7. Cell Scratch Assay

A 24-well plate was seeded with 4T1 cells. When the cell growth density reached 80%, a trace was made in each well. The scratched cells were washed with PBS three times. RMPI 1640 medium containing 2% FBS and 10 μg/mL BMVE was added to each well and cell migration was observed by microscopy at 12 and 24 h.

### 2.8. Cell Apoptosis

A 12-well plate was seeded with 4T1 or MCF-7 cells. Cells were treated with 20 and 40 μg/mL BMVE for 24 and 48 h. The cells were gathered, treated with the 7-AAD/Annexin V-APC Apoptosis Kit (KeyGEN BioTECH, Shanghai, China) and then analyzed by flow cytometry.

### 2.9. Live–Dead Assay

A 12-well plate was seeded with 4T1 cells. Cells were treated with 20 and 40 μg/mL BMVE for 24 and 48 h. After treating the cells with a calcein-AM/PI double-staining kit (KeyGEN BioTECH, Shanghai, China), the cells were observed using a fluorescence microscope.

### 2.10. Reactive Oxygen Species (ROS) Detection

Samples of 4T1 cells were treated with 20 and 40 μg/mL BMVE for 12 and 24 h. The cells were cultured with 2′,7′-dichloro-fluorescein diacetate (DCFH-DA) (Beyotime, Shanghai, China) for 30 min at 37 °C. Using fluorescence microscopy, the cellular ROS level was then determined.

### 2.11. Mitochondrial Membrane Potential Assay

A 24-well plate was seeded with 4T1 cells. Cells were treated with 20 and 40 μg/mL BMVE for 24 and 48 h. After treating the cells with a mitochondrial membrane potential kit (LEAGENE, Beijing, China) and DAPI, the cells were observed using a fluorescence microscope.

### 2.12. Assay of 3D-Cultured 4T1 Cells

Calcein-AM/PI double-staining assay: In 96-well U-shaped bottom plates, 4T1 cells were seeded in an RPMI 1640 medium containing 10% FBS and supplemented with 0.25% methylcellulose (1 × 10^4^ 4T1 cells in each well). The cells were incubated for 24 and 48 h with the addition of 20 and 40 μg/mL BMVE after spheroid formation. Then, the 3D spheroids were treated with a calcein-AM/PI double-staining kit and observed using a fluorescence microscope.

Morphological assay: 4T1 cells were resuspended in an RPMI 1640 medium containing 10% FBS and supplemented with 0.25% methylcellulose (5 × 10^4^ 4T1 cells in each well) and then seeded in 96-well U-shaped-bottom plates. After spheroid formation, 40 μg/mL BMVE was added to the experimental group and an equal volume of PBS to the control group for 48 h. Then, the 3D spheroids were observed using a microscope.

### 2.13. Animals and Tumor Model

Female BALB/c mice aged 4 to 6 weeks were obtained from the Shanghai Laboratory Animal Center. All mice were raised in an SPF-approved animal facility. All procedures were approved by the Animal Ethics Committee of Shanghai Jiao Tong University and carried out in accordance with the University’s Guidelines for Care and Use of Laboratory Animals. In total, 2 × 10^6^ 4T1 cells were injected into the mice’s flanks. Experiments were started when the mean volume reached 100 mm^3^. A total of 16 mice were used, and they were randomly divided into two groups of 8 mice each, named the control group and the experimental group. DiR dye or DiR-labeled BMVE was administered to each group of mice by tail vein or intratumoral injection, and fluorescence imaging was performed to observe the biodistribution of BMVE in the mice, as well as BMVE intratumoral injection to observe tumor treatment in vivo.

### 2.14. Hemolysis Assay

Blood was collected from the mice and sodium heparin solution was added to prevent blood clotting. The collected blood was centrifuged at 4500 rpm for 5 min. The supernatant was discarded and the red blood cells (RBCs) at the bottom of the centrifuge tube were resuspended in PBS and washed five times. The collected RBCs were resuspended in PBS. The negative control (RBCs in PBS), positive control (RBCs in water) and experimental groups with different BMVE concentrations (5, 10, 50, 100, 250, 400 μg/mL) were set up. The samples were allowed to stand at room temperature for 3 h and then centrifuged. Hemolysis results were estimated by a microplate reader (absorbance wavelength: 541 nm).

### 2.15. In Vivo Biodistribution of BMVE

Mice were injected with DiR dye or DiR-labeled BMVE via tail vein or intratumoral injection. At different points in time, the mice were imaged to observe whole-body fluorescence distribution. After 48 or 72 h, animals were sacrificed, and major organs were imaged.

### 2.16. In Vivo Anticancer Effects of BMVE

Tumors were injected intratumorally when the mean volume reached 100 mm^3^. Every other day for one week, mice were injected intratumorally with BMVE or an equal volume of PBS. After 15 days, the mice were sacrificed, and the tumors were weighed. Tumor sections were then histopathologically analyzed using hematoxylin and eosin (H&E) staining, terminal-deoxynucleotidyl transferase mediated dUTP-biotin nick end labeling (TUNEL) staining and Ki67 staining. H&E staining was performed on mouse major organs (heart, liver, spleen, lung and kidney).

### 2.17. Statistical Analysis

As indicated, the results are presented as mean ± standard deviation (mean ± SD) or mean ± standard error of the mean (mean ± SEM). The analysis was performed using a Student’s *t*-test. The threshold for a statistically significant difference was defined as * *p* < 0.05, ** *p* < 0.01 and *** *p* < 0.001.

## 3. Results and Discussion

### 3.1. Isolation and Cellular Internalization of BMVE

The experimental steps for isolating BMVE from bitter melon juice by centrifugation are depicted in Figure 1a. Insoluble impurities were removed from the bitter melon juice by two previous centrifugations, and the collected supernatant was almost transparent in color. The BMVE can be separated from other soluble impurities in the supernatant by adding Opti-prep and ultracentrifugation. The concentration and diameter of isolated BMVE was determined by NTA (Figure 1b), and the diameter was primarily centered at 147 nm. Following further characterization by TEM, BMVE showed a typical “cup-shaped” morphology and integrity (Figure 1c and Appendix A). These results suggested BMVE had been successfully isolated from bitter melon.

The antitumor effects of BMVE are based on their capacity to be internalized by tumor cells. We first examined the degree of BMVE internalization by 4T1 cells. BMVE was labeled with the lipophilic dye Dil (1,1′-Dioctadecyl-3,3,3′,3′-tetramethylindocarbocyanine perchlorate). Samples of 4T1 cells were treated with Dil-labeled BMVE for various durations at 37 °C, then the nuclei were stained with DAPI and visualized using confocal microscopy (Figure 1d). At six hours, the results indicated that 4T1 cells had begun to internalize BMVE. The internalization of BMVE by 4T1 cells was time-dependent, and the degree of internalization increased with time at a constant BMVE concentration. Moreover, we analyzed the degree of internalization of 4T1 cells with different BMVE concentrations at the same time point by flow cytometry (Appendix A). These results demonstrate the effective cellular uptake of BMVE. Enhanced internalization may represent enhanced antitumor effects; thus, increasing BMVE concentration and treatment time is likely to increase the anti-proliferative and apoptosis-inducing effects.

### 3.2. Anti-Proliferation and Migration Inhibition Effects of BMVE

We observed that BMVE had anti-proliferative effects on breast cancer cells. The CCK-8 assay revealed that BMVE was able to inhibit the proliferation of 4T1 and MCF-7 cells (Figure 2a and Appendix A), and the effect could be seen 12 h after treatment. Higher concentrations increased the anti-proliferative effect of BMVE during the same treatment period. Moreover, the anti-proliferative effect at the same BMVE concentration increased over time. Consequently, the anti-proliferative effect of BMVE on breast cancer cells is dependent on time and BMVE concentration. Low BMVE concentration with prolonged treatment can give the same anti-proliferation results as high BMVE concentration with short time treatment in a certain range. Experiments on colony formation also confirmed that BMVE could inhibit the growth of 4T1 cells (Figure 2b,c). Thus, the antitumor activity of BMVE is at least partly due to its anti-proliferative effect. On the other hand, inhibition of tumor cell migration is also an aspect for antitumor effect. In order to observe whether BMVE has the ability to inhibit 4T1 tumor cell migration, we performed a cell scratch assay. The experimental results showed that the migration percentage of the BMVE-treated group was lower than the control group at 12 and 24 h, which indicated that BMVE had the ability to inhibit the migration of 4T1 cells (Figure 2d,e). These results validate that BMVE can serve as a novel therapeutic agent for breast cancer therapy.

### 3.3. Induction of Apoptosis

Furthermore, we evaluated the cell apoptosis induced by BMVE. Samples of 4T1 cells were incubated with varying concentrations of BMVE for 24 or 48 h (Figure 3a,b). Flow cytometry results indicated approximately 21% apoptosis at 20 μg/mL BMVE treated for 24 h. Increasing the concentration and incubation time of BMVE treatment on 4T1 cells could induce apoptosis to a greater extent. Samples of 4T1 cells treated with 40 μg/mL BMVE for 48 h could undergo 55% apoptosis. We also observed a similar trend of apoptosis in MCF-7 breast cancer cells (Appendix A). By double staining cells with calcein-AM/PI, cell survival (green) and cell death (red) were further observed (Appendix A). After treated with BMVE, the survival of 4T1 cells was significantly diminished.

The apoptosis-inducing effect of BMVE is probably related to the generation of intracellular reactive oxygen species (ROS) [21]. By measuring the fluorescence intensity of DCFH-DA (2′,7′–dichlorofluorescin diacetate), the intracellular ROS level in 4T1 cells was determined. Fluorescence microscopy revealed that the intracellular fluorescence intensity was significantly increased after treated by BMVE and has a time- and concentration-dependent trend, indicating that BMVE could significantly induce ROS production in 4T1 cells (Figure 3c,d). We then analyzed the ROS production in MCF-10A mammary epithelial cells under the same treatment conditions. The level of ROS in MCF-10A cells after BMVE treatment was clearly lower than that in 4T1 cells (Appendix A).

Studies have shown that the production of ROS causes damage to mitochondria [24]. Mitochondrial membrane potential (ΔΨm) is a crucial indicator for assessing mitochondrial function and is associated with early cell apoptosis [25]. A decrease in mitochondrial membrane potential is a marker of early apoptosis. We characterized the ΔΨm by JC-1 (5,5′,6,6′-tetrachloro-1,1′,3,3′-tetraethyl-imidacarbocyanine iodide) assay. Samples of 4T1 cells were treated with 20 and 40 μg/mL BMVE for 24 and 48 h, respectively, and the fluorescence levels were observed by fluorescence microscopy after staining with JC-1 (Figure 3e,f). JC-1 can aggregate in the mitochondrial matrix and produce red fluorescence at high ΔΨm. Conversely, it is present as a monomer and exhibits green fluorescence. These results demonstrated that the intracellular ΔΨm of 4T1 cells were decreased after BMVE treatment compared with the control group, which also showed a BMVE concentration- and treatment time-dependent trend, consistent with the trend of intracellular ROS generation and apoptosis. Thus, the ability of BMVE to induce apoptosis in tumor cells is ascribed to the generation of ROS and the decrease in ΔΨm, which are also essential aspects for antitumor effects.

### 3.4. Effect of BMVE on 3D-Cultured 4T1 Cells

We obtained 3D-cultured 4T1 cells using round-bottom 96-well plates and complete medium supplemented with methylcellulose. These 3D-cultured 4T1 cells were created to mimic tumors in vitro [26]. The degree of apoptosis and morphological changes of the cell spheroids in the presence or absence of BMVE were observed in the same environment to determine the role they might play in solid tumors. Firstly, we treated 3D-cultured 4T1 cells with 20 and 40 μg/mL BMVE and carried out calcein-AM/PI analysis. This result demonstrated a significant increase in the proportion of dead cells (red fluorescence) in 3D cell spheroids after BMVE treatment (Figure 4a,b). Furthermore, we used larger diameter 3D cell spheroids to see more clearly whether BMVE affects their morphological changes. We found that the 3D cell spheroids treated with BMVE became looser and clearly larger in diameter (Figure 4c,d). This was probably due to both anti-proliferative and apoptosis-inducing effects of BMVE on 4T1 cells. According to these results, we believe that BMVE can effectively restrain the growth of breast cancer in vivo.

### 3.5. In Vivo Biodistribution and Anticancer Effects of BMVE

To assess the biodistribution and antitumor effect of BMVE in vivo, a 4T1 tumor-bearing mouse model was developed. We first performed a hemolysis assay before starting the in vivo applications. As shown in Appendix A, no hemolysis occurred in red blood cells (RBCs) treated with a high concentration of 400 μg/mL BMVE. BMVE exhibited good hemocompatibility; thus, we proceeded to perform further in vivo experiments. Mice were injected with free DiR dye (control) or DiR-labeled BMVE via a tail vein. The distribution of BMVE in vivo was imaged at various time points (Figure 5a). After intravenous injection, most of the free DiR dye was quickly cleared in the body, while DiR-labeled BMVE was predominantly distributed in the liver (Figure 5a). The ex vivo fluorescence images also revealed that free DiR dye was mainly metabolized by the kidneys and DiR-labeled BMVE was mainly accumulated at the liver with certain tumor targeting (Figure 5b,c). The tumor-targeting capacity of BMVE can probably be attributed to the enhanced permeability and retention (EPR) effect [27]. Moreover, we investigated the retention time of BMVE at the tumor site. After intratumoral injection, the fluorescence signal of DiR-labeled BMVE could be maintained at a relatively high level after 72 h (Figure 5d,e). On the contrary, the fluorescence intensity of the free DiR dye gradually diminished after 4 h of injection. The ex vivo fluorescence images also revealed the long retention ability of BMVE at tumor site (Figure 5f,g). As mentioned above, given a longer incubation time of BMVE with 4T1 cells, a stronger antitumor effect could be achieved. Therefore, the long retention of BMVE at the tumor site can contribute to more efficient tumor suppression.

Encouraged by the excellent results above, we then evaluated the therapeutic effect of BMVE on 4T1 tumor-bearing mice. The in vivo therapeutic process is shown in Figure 6a. When the tumor volume reached 100 mm^3^, PBS (control) or BMVE was injected intratumorally every other day during the first week for a total of four injections. After BMVE treatment, the tumor volume (Figure 6b) and weight (Figure 6c,d) were significantly reduced compared with the control group. The inhibition rate of the BMVE group was calculated to be 56.4%. Furthermore, tumors were obtained for hematoxylin and eosin (H&E) staining, terminal-deoxynucleotidyl transferase mediated dUTP-biotin nick end labeling (TUNEL) staining and Ki67 immunohistochemical assays to confirm the antitumor effect of the treatments (Figure 6e). As shown in H&E staining, the BMVE group displayed severe nucleus shrinkage, karyorrhexis, demonstrating dramatic tumor apoptosis and necrosis. Furthermore, TUNEL staining showed that the BMVE group had more DNA breaks in their nuclei, indicating a higher proportion of apoptotic cells. Ki67 immunohistochemical detection was also performed to assess the proliferative capacity of the tumor. As depicted in Ki67 detection, the proliferation of tumor cells which had been treated with BMVE was obviously restrained. There was no significant difference in body weight after BMVE treatment (Figure 6f). In addition, the major organs of mice (i.e., heart, liver, spleen, lung and kidney) after treatment were collected for H&E staining. As shown in Figure 6g, no visible damage or inflammation was observed in the BMVE group. All of above results indicate that BMVE has outstanding biosafety and histocompatibility and can exert a superior antitumor effect without side effects.

## 4. Conclusions

BMVE was isolated using ultracentrifugation in this study. The vesicles were determined to have a typical “cup-shaped” morphology by TEM, and the particle size measured by NTA was primarily concentrated at 147 nm. BMVE could be internalized by 4T1 cells in a concentration- and time-dependent manner, according to in vitro cell experiments. CCK-8 analysis and colony formation assays validated the anti-proliferative effect of BMVE on breast cancer cells. The anti-proliferative effect of plant-derived vesicles extracts (PDVEs) on tumor cells is closely related to the induction of tumor cell cycle arrest, which is associated with a decrease in cell cycle protein level [22,28]. Moreover, cell scratch assays confirmed the ability of BMVE to inhibit the migration of tumor cells. The effect of migration inhibition was partially attributed to matrix metalloproteinase 9 (MMP9), a migration-associated protein [22].

Then, we confirmed that the ability of BMVE to induce cell apoptosis was ascribed to the generation of intracellular ROS. Yang et al. [21] revealed that ROS production, induced by BMVE, played a positive role in tumor cell apoptosis. The upregulation of JUN, an important molecule related to the MAPK signaling pathway [29,30], was a potential molecular mechanism of ROS production. ROS can damage mitochondria, which is reflected by the decrease in intracellular ΔΨm. In addition, PDVEs are composed of lipids, proteins, nucleic acids and metabolites. The antitumor activity of PDVEs is probably related to these biochemical molecules [31]. For instance, Wang et al. identified 81 microRNAs (miRNAs) enriched in BMVE by RNA sequencing. Multiple miRNAs are involved in the regulation of the PI3K/AKT signaling pathway to achieve antitumor activity [22]. The presence of other RNA types should also be investigated in the future. Furthermore, the metabolites in bitter melon, which were analyzed in previous studies, are mainly classified as cucurbitane triterpenes, cucurbitane triterpene glycosides, phenolic acids, flavonoids, essential oils, fatty acids, amino acids, sterols and saponins [16]. It is possible to further investigate which metabolites are contained in BMVE on this basis. The results of lipid and protein analysis will also provide further insight into the potential antitumor mechanisms of BMVE.

The calcein-AM/PI double staining and morphological changes of 3D-cultured 4T1 cells further reflected the potential antitumor capacity of BMVE to destroy solid tumors. In vivo imaging revealed that BMVE was primarily metabolized through the liver and possessed a partial tumor-targeting ability. More importantly, BMVE could remain at the tumor site for longer than 72 h. This indicated the longer retention of BMVE in vivo, which is expected to be used as a carrier for sustained drug release. The biosafety and biocompatibility of BMVE were verified. Finally, in vivo therapeutic experiments revealed that BMVE could effectively inhibit tumor growth without noticeable side effects.

As a potential natural nanomedicine for cancer treatment, BMVE exhibits a robust antitumor effect on breast cancer cells in vitro and in vivo without obvious side effects, which increases the future interest and application of PDVEs derived from medicinal and edible plants to treat tumors or other diseases. On this basis, the delivery potential of BMVE can be exploited, and one or more drugs can be loaded to achieve combination therapy. BMVE is anticipated to play a more significant role in tumor therapy if it is modified or transformed to enhance the vesicles’ targeting capability.

## Data Availability

The data presented in this study are available in this article and Appendix A.

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
