# Peer review of "Anticancer Activity of Bitter Melon-Derived Vesicles Extract against Breast Cancer"

_cells, 2023, doi:10.3390/cells12060824_

Round 1

Reviewer 1 Report

The objective of this study is to elucidate the effect of the bitter melon-derived vesicles extract (BMVE) on antitumor activity against breast cancer cells using 4T1 murine mammary tumor cells. The authors isolated BMVE from bitter melon juice and demonstrated that BMVE exhibited inhibitory effects on the proliferation and migration of 4T1 cells and induced apoptosis, probably through the generation of intracellular ROS. The authors also showed the biodistribution of BMVE in mice and in vivo therapeutic efficacy using a syngenic xenograft tumor model. They conclude that BMVE has antitumor effects in vitro and in vivo without obvious side effects.

The conclusions are well supported by in vitro and in vivo experiments. In my opinion, the manuscript could be suitable for publication in Cells after the authors address the following issues.

 major points:

(1) 4T1 is a murine mammary tumor cell line. The effect of BMVE on anti-proliferation and induction of apoptosis should be conducted using more than one human breast cancer cell line.

(2) BMVE has no side effects in various organs in a mouse model. Does BMVE generate ROS only in tumor cells? The authors should determine the effect of BMVE on the generation of ROS in non-transformed normal mammary epithelial cells

 minor points:

(1) It would be useful to discuss the contents of BMVE and the proposed molecular mechanism of how BMVE inhibits the proliferation and migration of tumor cells.

(2) The authors should discuss how BMVE generates ROS in 4T1 cells. 

Reviewer 2 Report

Dear Dr Liu and coworkers,

It is interesting to read about the application of these extracts as cancer treatments. As an organic chemist, I am wondering about the secondary metabolites involved in bioactivity and I would like to read about them. If you can share this information or add it to the manuscript it would be helpful. Also, could explain further investigation.

  • The paper is the evolution of any previously published paper. If yes, please add some background of demonstrated bioactivity (i.e.: DOI:10.1186/s12951-021-00995-1).
  • The manuscript is well written even though a revision by a mother tongue speaker is highly advisable (Many typos and mistakes to be corrected before publication)

Some are listed below:

Please introduce a description for any acronym used in the paper. i.e.: ROS (add in line 130), RBCs, PDVEs

Line 283: methylcellulos correct to methylcellulose

Use capitals and low case in case of consider necessary (own names, etc): Calcein-AM/PI correct to calcein-AM/PI

Please consider improving the English in the conclusion section (4): Some ideas were presented using informal English. Avoid the use of contractions (i.e.: line 365).

Is the supporting information reported in the main text?

Yours sincerely,

Guillermo A. Guerrero Vasquez

Round 2

Reviewer 1 Report

The authors answered to all my comments well. I do not have any more questions.